# The representation of object drawings and sketches in deep convolutional neural networks

**Johannes Singer**
Vision and Computational Cognition Group
Max Planck Institute for Human Cognitive and Brain Sciences
Leipzig, Germany
Department of Psychology, Ludwig-Maximilians-Universität
Munich, Germany
jsinger@cbs.mpg.de

**Katja Seeliger**
Vision and Computational Cognition Group
Max Planck Institute for Human Cognitive and Brain Sciences
Leipzig, Germany
katjamueller@cbs.mpg.de

**Martin N. Hebart**
Vision and Computational Cognition Group
Max Planck Institute for Human Cognitive and Brain Sciences
Leipzig, Germany
hebart@cbs.mpg.de

## Abstract

Drawings are universal in human culture and serve as tools to efficiently convey meaning with little visual information. Humans are adept at recognizing even highly abstracted drawings of objects, and their visual system has been shown to respond similarly to different object depictions. Yet, the processing of object drawings in deep convolutional neural networks (CNNs) has yielded conflicting results. While CNNs have been shown to perform poorly on drawings, there is evidence that representations in CNNs are similar for object photographs and drawings. Here, we resolve these disparate findings by probing the generalization ability of a CNN trained on natural object images for a set of photos, drawings and sketches of the same objects, with each depiction representing a different level of abstraction. We demonstrate that despite poor classification performance on drawings and sketches, the network exhibits a similar representational structure across levels of abstraction in intermediate layers which, however, disappears in later layers. Further, we show that a texture bias found in CNNs contributes both to the poor classification performance for drawings and the dissimilar representational structure, specifically in the later layers of the network. By finetuning only those layers on a database of object drawings, we show that features in early and intermediate layers learned on natural object photographs are indeed sufficient for downstream recognition of drawings. Our findings reconcile previous investigations on the generalization ability of CNNs for drawings and reveal both opportunities and limitations of CNNs as models for the representation and recognition of drawings and sketches.

2nd Workshop on Shared Visual Representations in Human and Machine Intelligence (SVRHM), NeurIPS 2020.

# 1 Introduction

From ancient drawings on cave walls to sophisticated graphs on digital displays, humans have used different forms of media to convey information in visual form. A simple and highly efficient way of visualization is provided by line drawings. For objects, line drawings abstract away from details such as color and texture and sometimes even distort object parts. Yet, humans effortlessly recognize them as the physical objects they are meant to depict [1, 2]. This behavior suggests that despite their simplicity, line drawings capture many of the core visual features that are essential for the recognition of real-world objects. This view is supported by neuroimaging studies demonstrating that similar brain regions are involved in the recognition of objects and line drawings [3]. As a computational model of object recognition, deep convolutional neural networks trained on large databases of natural images have recently gained popularity in visual neuroscience [4], due to their - at times - close correspondence to brain activity in visually-responsive regions. The apparent similarities between processing of photographs and drawings of objects in the human brain and the parallels between CNNs and the visual brain responses together suggest that CNNs might recognize drawings similarly to humans. Demonstrating such similarities would provide a mechanistic explanation for the natural emergence of visual abstraction abilities, while at the same time opening the avenue towards in silico study of the representation of drawings.

Previous work aimed at quantifying the generalization ability of CNNs to drawings has led to conflicting findings. On the one hand, CNNs such as AlexNet have yielded poor classification performance for abstract drawings [5]. This result is supported by evidence suggesting that, in contrast to humans, CNNs focus less on shape but more strongly on features such as texture [6] which are different for drawings and natural images. On the other hand, others have argued that the representations in CNNs, as measured with representational similarity analysis [7], are highly similar for photographs and abstract drawings [8]. This is in line with findings showing that the representation of shape information, which in large parts is common in photographs and drawings, is similar in CNNs and humans [9]. These divergent interpretations point to a gap in our understanding of the generalization ability of CNNs to drawings. Here, we aim to close this gap by examining the processing of a CNN trained on natural images on a set of photos, drawings and sketches of the same objects, reflecting different levels of abstraction. Importantly, we do not only focus on the classification performance of the network but also quantify generalization in terms of the network's internal representational similarity, thus providing a more complete picture of the network's behavior for object images across different levels of abstraction.

# 2 Methods and Results

**Experiment 1**   We used the VGG-16 architecture [10] trained on the ILSVRC2012 dataset [11] for our experiments, which shows well-documented similarities to visual object processing in the human brain [12, 13]. For simplicity, we will refer to networks trained with this dataset as "ImageNet-trained". To probe the generalization ability of the network, we used a set of 42 object images in three different depictions (Fig 1): (1) Natural photographs of objects cropped from their background, (2) the exact same objects but as line drawings, removing color and most of the surface texture, and (3) abstract sketches of these objects, reducing drawings even further and leaving highly simplified objects including changes in visual appearance. The drawings and sketches were created for the purpose of this study, and images were not derived from any of the datasets that were used for training the networks in our experiments. Using the same objects in different depictions allowed us to directly compare how a given network responds to object images across different levels of abstraction.

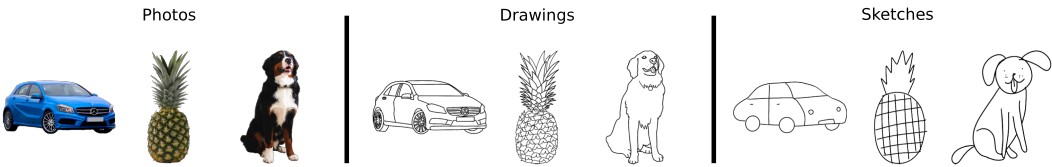

Figure 1: **Example stimuli from the stimulus set used in all experiments.** The set contained the same 42 object images as photos, drawings and sketches, representing different levels of abstraction.

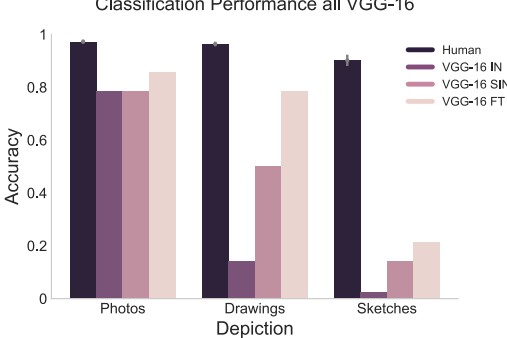

Figure 2: **Classification performance of humans and variants of VGG-16.** Humans recognized photos, drawings, and sketches very well, with only slightly decreased performance for sketches. VGG-16 trained on ImageNet (VGG-16 IN) performed well on photos but showed poor performance for drawings and sketches. VGG-16 trained on stylized ImageNet (VGG-16 SIN) showed improved performance for drawings but not for sketches. VGG-16 finetuned on ImageNet-Sketch (VGG-16 FT) largely restored performance for drawings, yet performance for sketches remained very weak.

As a first step, we probed the classification performance of an ImageNet-trained VGG-16 (VGG-16 IN) on each of the depictions separately and compared it to human performance. For this purpose, we passed the images through the network and compared the highest scoring label in the softmax layer with the true object label. We scored a response as correct if the label predicted by the network matched the true category label of the image or any of its hyponyms in the WordNet hierarchy [14]. Human performance was measured in an online experiment by asking participants (N=480) to label the object images used in this experiment (all participants provided their informed consent and the experiment was approved by the local ethics committee). Importantly, a given participant always rated images of only one depiction, making sure there was no carry-over effect between levels of abstraction. We found that the network performed well on photos but showed poor performance both for drawings and sketches (Fig 2, VGG-16 IN). Human performance, on the other hand, was close to ceiling for all three depictions, showing that humans recognized photos and drawings similarly well, with only a slight decrease in performance for sketches. Together, these results demonstrate that, in contrast to humans, an ImageNet-trained VGG-16 generalizes poorly to drawings and sketches in terms of classification performance.

While classification accuracy informs about the generalization ability in a network's overt behavior, it leaves open the possibility of similar representational structure across levels of abstraction in its latent representations. To quantify the degree of representational similarity between photos, drawings, and sketches across different processing stages in the CNN, we applied representational similarity analysis (RSA) [7]. RSA compares representational dissimilarity matrices (RDMs) of all pairs of activation patterns, thus measuring the relative dissimilarity of all objects to each other and allowing us to compare this dissimilarity structure across depictions. We computed RDMs for every depiction and every layer in the network separately using the Pearson correlation distance as a dissimilarity metric $(1 - r)$. For our analyses, we focused on the pooling layers and the fully connected layers. Subsequently, we compared all three RDMs for a given layer, using the Spearman rank correlation coefficient. In the RSA framework, a high correlation between RDMs of different depictions would be interpreted as a representational format that is shared across different levels of abstraction.

In contrast to the low classification accuracy for drawings, we found a surprisingly high degree of similarity between the photo and drawing representations in intermediate layers (Fig 3a). The similarity, however, dropped sharply after pooling layer 4. A similar pattern was observed for the photo-to-sketch similarity, however overall weaker. The drawing-to-sketch similarity, on the other hand, increased steadily up to the penultimate layer. Hence, even though the network performed poorly for drawings and sketches, there were high similarities in how it represented photos and drawings of objects. Yet, this similarity disappeared towards the later layers of the network.

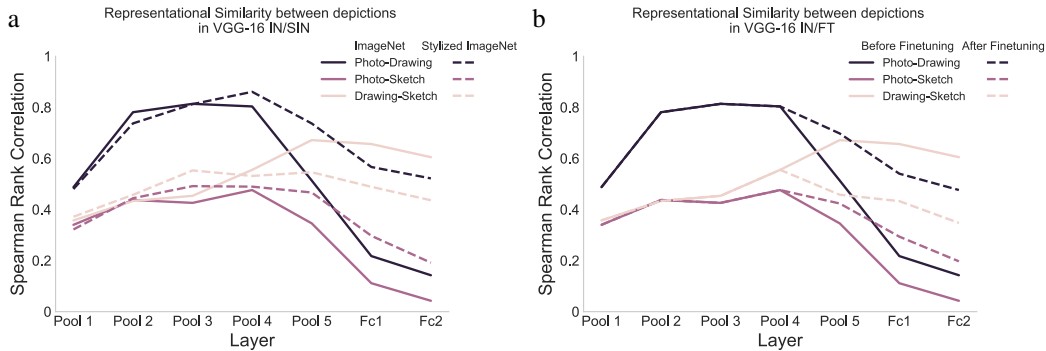

Figure 3: **Representational similarity in variants of VGG-16.** a) VGG-16 SIN showed a similar pattern of representational similarity for photos and abstracted depictions as VGG-16 IN in the layers up to pooling layer 4. However, in the late layers representational similarity was higher in VGG-16 SIN. This suggests that texture bias contributes to the low similarity across levels of abstraction observed in VGG-16 IN. b) After finetuning, the representational similarity between photos and abstracted depictions was increased for all finetuned layers. Together with the classification results this demonstrates that the representations in early and intermediate layers of an ImageNet-trained CNN are already sufficiently domain-general for the categorization of drawings.

**Experiment 2**    A simple explanation for the poor classification performance observed in the CNN in Experiment 1 could be the texture bias previously described for ImageNet-trained CNNs. In contrast to humans who primarily rely on shape for the recognition of objects, it has been shown, that in the case of conflicting shape and texture information, CNNs show a bias towards texture [6, 15]. Since texture is strongly altered in drawings and sketches compared to the images the network was trained on, the focus on texture might bias the underlying representations in the network, leading to incorrect decisions at the classification stage. To test whether attenuating the texture bias would generally increase the similarity in processing of photos, drawings and sketches in terms of both classification performance and representational similarity, we probed the generalization ability in a shape-biased CNN trained on stylized ImageNet on the same images as in Experiment 1.

VGG-16 trained on stylized ImageNet (VGG-16 SIN) indeed showed better classification performance on drawings compared to its ImageNet-trained counterpart (VGG-16 IN) (Fig 2). However, no such improvement was found for highly-abstracted sketches. Thus, attenuating the texture bias led to better generalization for drawings but not for sketches in terms of the network's classification. The representational similarity for photos and drawings was increased for VGG-16 SIN relative to VGG-16 IN. This increase was limited to later layers, where similarity was largely restored. Yet, training on stylized ImageNet led to no increase in representational similarity in the early and intermediate layers. In addition, the drawing-to-sketch similarity was reduced in the late layers of VGG-16 SIN, indicating a reduction of the bias that contributes to their similarity in VGG-16 IN.

Together, the results suggest that texture bias contributes to the poor generalization ability to drawings and sketches we found in the ImageNet-trained CNN. However, the degree of representational similarity found up to intermediate layers is not altered when using a shape-biased CNN. This, in turn, raises the question whether representations in the intermediate layers of an ImageNet-trained CNN are already sufficiently general to support the recognition of drawings and possibly sketches. Under this view, a domain-related bias towards photographs might prevent the network from capitalizing on this shared representational format.

**Experiment 3**    To directly test the hypothesis that features learned on natural images in early and intermediate layers in the network enable representations that are general enough to lay the foundation for the recognition of drawings and sketches, we finetuned the network on the ImageNet-Sketch dataset [16]. This dataset contains approximately 50 drawing images, gathered from Google image search, for each of the 1000 ImageNet-classes VGG-16 had been trained on. We reasoned that if the features in early and intermediate layers are general enough to support recognition of drawings and sketches, they should easily transfer to the task of categorizing these images. Therefore, we used the

ImageNet-trained VGG-16 from Exp 1, froze the weights of all the layers up to pooling layer 4, and finetuned all the other layers starting from layer conv5-1 on the ImageNet-Sketch dataset.

After finetuning, the classification performance on drawings and photos was similarly high, reflecting a restoration of classification ability for drawings in comparison to VGG-16 IN (Fig 2). Yet, sketch performance remained poor and was not better than in the other networks. In sum, VGG-16 FT generalized very well to drawings but still did not generalize well to sketches.

Comparing the representational similarities between depictions in VGG-16 IN and VGG-16 FT, we found that the finetuning procedure significantly increased the representational similarity for photos and abstracted depictions and decreased the similarity for drawings and sketches (Fig 3b). With respect to VGG-16 SIN, there were no significant differences in the representational similarities between depictions in the finetuned layers. Taken together, after finetuning the network exhibited a more similar representational format across levels of abstraction akin to the results in VGG-16 SIN. This suggests that keeping features in early and intermediate CNN layers that had been learned on natural images allowed for representational structure across depictions comparable to a shape-biased CNN.

In conclusion, we show that features in early and intermediate layers learned on photographs are useful for categorizing drawings. However, in the process of linking them to high-level features and classification decisions, a domain-related bias arises which leads to a dissimilar representational format across levels of abstraction and prevents correct classifications for drawings and sketches.

## 3 Discussion

The aim of this study was to reveal the extent to which CNNs process photos, drawings and sketches in a similar way, a characteristic that has been shown repeatedly in humans [1, 2, 3]. We demonstrate that intermediate layers of the network show strong similarities in the representational structure of object images across levels of abstraction. However, these similarities do not extend into the later stages of the network and result in poor classification performance for drawings and sketches. We identify that the texture bias present in ImageNet-trained CNNs [6] is a contributing factor, specifically towards the similarity structure of representations in later layers and the ensuing classification performance of the network. Finally, by restoring the network's performance for drawings through finetuning of later layers, we provide direct evidence that the features in early and intermediate CNN layers learned from natural images are also useful for the categorization of drawings. This suggests that the ability to recognize abstract drawings may in part be an emergent property of networks optimized towards the goal of object recognition in natural images. These findings resolve apparent inconsistencies in previous studies [5, 8] by providing a more complete picture of the generalization to drawings, both in the network's representational similarity and its classification performance. Despite these results, the very limited generalization to highly simplified sketches in all our experiments illustrates the limitations of the generalization ability of ImageNet-trained CNNs with similar architecture and suggests major differences to recognition strategies employed by humans.

One intriguing possibility for the limited generalization of CNNs may be related to the architecture of the network used in this study. Indeed, a drop in representational similarity was found mostly for fully-connected layers, and previous work has highlighted differences to the human brain for this type of layer [12]. In future work, it would be possible to investigate the degree to which these results hold true also for different CNN architectures. Regardless, fully-connected layers are an integral part of many currently-used CNN architectures. What this study shows is that a common CNN architecture exhibits sufficiently general intermediate representations but suffers from limitations in generalization to abstracted depictions that can be overcome partially by changes in training.

The generalization of CNNs from photos to abstract sketches could not be restored with our alternative training approaches. This indicates that the recognition of sketches might require different strategies than the recognition of natural images and drawings. A key factor for robust recognition may lie in the ability to represent global shape, which has been shown to be limited for currently predominant CNN architectures [15, 17]. However, it is possible that finetuning with more abstract sketches would lead to similar generalization performance. Future investigations into the brain mechanisms that enable the recognition of abstract sketches and innovations in computational models of visual processing might provide mechanistic explanations for the robust recognition of such highly abstracted objects images that is found in humans.

In conclusion, our results provide a deeper understanding of the representation of drawings and sketches in CNNs, by harmonizing seemingly inconsistent findings, revealing the effects of the texture bias on drawing representations, demonstrating the usefulness of features in intermediate CNN layers for the recognition of drawings, and highlighting limitations of ImageNet-trained CNNs as general models for the representation of drawings and sketches.

## Broader Impact

A major obstacle for the real-world application of current CNN architectures is their limited out-of-sample generalization ability [18]. Our approach lends support to recent efforts quantifying the generalization ability of CNNs in terms of both their internal representations and their classification performance [19, 20] and provides a framework for a deeper understanding of how networks trained on natural images represent and classify images from other domains. While our work does not directly aim at providing new real-world applications, our results offer a mechanistic account for the natural emergence of visual abstraction ability in a system optimized for recognition of natural object images, however, suggesting different mechanisms for very high levels of abstraction. Finally, our study shows the opportunities and the limitations of CNNs as computational models of the representation of drawings.

## Acknowledgments and Disclosure of Funding

We thank Tim Kietzmann for helpful comments on an earlier version of the manuscript. This work was supported by a Max Planck Research Group grant of the Max Planck Society awarded to MNH. The authors declare no conflict of interest.

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
