# OpenReview forum: "The representation of object drawings and sketches in deep convolutional neural networks"
_NeurIPS.cc/2020/Workshop/SVRHM — SVRHM@NeurIPS Poster_

### Official Review · AnonReviewer1 · 2020-10-23
**Deepens understanding of how CNNs interpret drawings and sketches**

**Rating:** 8
**Confidence:** 4

**Review:**

This paper develops a series of experiments to further our understanding of how convolutional neural networks interpret sketches and drawings of objects after being trained on real world photos. The authors proceed through a very logical sense of experiments. These start with the selection of the dataset itself, and the authors delineation of "drawings" and "sketches" was insightful and not something I believe I've seen done before in this literature - and the results later showed that this separation was crucial to understanding the results (hopefully the authors will make this dataset openly available for others). After examining initial results reproducing poor classification of sketches and drawings, they proceed to demonstrate that classification on drawings can be improved by purely fine tuning later layers, and this is motivated by noticing higher representational similarity in the earlier layers of the network using representational similarity analysis.

Overall the arguments were presented well and the experimental technique was sound and supported the findings. My main criticism is that the results might be overfitting somewhat to the now dated VGG16 architecture - it has several peculiarities relative to other CNN architectures and so it would be interesting to do followup work confirming these results on at least one other neural backbone (however, the authors to their credit bring up this point themselves in paragraph 2 of the discussion).

My main other blind spot as a reviewer is my unfamiliarity with the mechanics of Representational Similarity Analysis (RSA) itself, which serves as a crucial building block of this analysis. This did not affect my rating, but did diminish my confidence measure provided.

Overall a very coherent paper well explained and supported by experiments, relevant to the workshop, and that furthers understanding of the topic and also leads to new related questions.

---

### Official Review · AnonReviewer3 · 2020-10-28

**Rating:** 7
**Confidence:** 4

**Review:**

1. Summary

The paper investigates the internal representations of natural images, line drawings and sketches in DNNs. It finds that while VGG16 performs very badly on the classification of drawings and sketches the representations especially in the mid levels are very similar. They compare these properties for ImageNet trained models, models with a shape bias and models fine tuned on sketches finding that the shape biased models show very similar properties to the fine tuned models.

2. Strengths/Weaknesses
+ Interesting problem
+ Simple but insightful analysis
+ Well selected experiments and comparison
+ Well done figures
- It would have been great to also see performance and representations if the whole model is finetuned on sketches or trained from scratch on sketches.
- The experiments are mostly on the surface and deeper analysis is somewhat missing. However for a workshop submission the level of analysis and detail is absolutely sufficient allowing to catch the meat of what the analysis of drawings and sketches may teach us about DNNs.


3. Recommendation

I recommend accepting the paper as it shows some very interesting first experiments analyzing DNNs based on the differences between the internal representations and classification performance of DNNs presented with photos, drawings and sketches.

4. Comments
- Do I understand figure 2 correctly that VGG finetuned on sketches performs better on the images of cropped natural objects than the ImageNet trained baseline?

---

### Official Review · AnonReviewer2 · 2020-11-02
**Very Clearly Written, Results are Somewhat Expected.**

**Rating:** 5
**Confidence:** 4

**Review:**

The paper has very good clarity and I appreciate the effort!

In this paper, the authors demonstrate:1) the poor generalization ability from CNNs trained on Imagenet to drawing and sketch data. (This is somewhat expected) 2) the lower and intermediate layers representation similarity of CNN-IN on photos, drawing, and sketch. 3) dissimilarity of CNN-IN on photos, drawing, and sketch.

Since humans can perform almost equally well on all three modalities listed above, a major hypothesis in this paper is that the representation from the intermediate layer of CNN-IN might be sufficient for drawing and sketch. To test this hypothesis, the author finetunes the higher layers of CNN-IN with Imagenet SIN and show the performance improve significantly on drawing data. (This is also somewhat expected since the finetuned portion is still quite large.) One very closely related result is that in [1], it was shown that if the second 4096D fully-connected layer output from the imagenet-trained Alexnet is used as a feature extractor, by using an SVM to finetune, the performance was around 67%. Since the authors are using VGG-16, I would expect the corresponding result would be even better. So an important ablation study is:
(1) Taking the output from the intermediate layer of VGG1`6-In and finetune with an SVM.
(2) Taking the output from a higher layer of VGG16-In and finetune with an SVM.

The performance of VGG16-FT still performs poorly on sketch data is also somewhat expected since drawing data is geometrically precise whereas sketches are much more abstract. The network has never been shown any sketches, even in an unsupervised manner. So I'm not sure what I can learn from this result. One way to generalize beyond drawing to sketch is to use augmentation to mimic the abstractions introduced to sketches. E.g. the local and structure deformation used in[2] and the dilation and erosion operator used in [3]. I'm not sure if these augmentation techniques would help the author in any way to improve the argument but they should help the generalization.

Overall, I like the paper and I think my score shall be increased if more solid ablation studies can be provided to support the argument.

[1] Sketch-a-Net that Beats Humans, BMVC15'
[2] A deep neural network that beat humans, IJCV17'
[3] Deep plastic surgery: Robust and controllable image editing with human-drawn sketches, ECCV20'

---

### Public Comment · ~Johannes_Singer2 · 2020-11-27
**Response to reviewers**

Dear reviewers,

We would like to thank you for providing insightful and constructive feedback on our submission.
We were pleased to see that all of you showed interest in our work and provided interesting follow up ideas on our analyses.
We are currently working on implementing follow-up analyses and will likely include further investigations in future work. In the camera ready version of the submission, we chose not to include these analyses, as we perceive the suggestions as extensions not critically affecting the conclusions drawn in our work. We hope this meets with your approval.

Thank you again for your time and the feedback!

---

### Decision · Program_Chairs · 2020-11-02

Accept (Poster)